# Effect of Particle Size of Silage of Flint Corn Grain on Dairy Cows Fed Tropical Pasture: Performance, Intake, Ruminal Fermentation, and Digestibility

**DOI:** 10.3390/ani13121932

**Published:** 2023-06-09

**Authors:** Débora R. Gomide, Renata A. N. Pereira, Rayana B. Silva, Josué T. R. Carvalho, Márcio A. S. Lara, Marcos N. Pereira

**Affiliations:** 1Departamento de Zootecnia, Universidade Federal de Lavras, Lavras 37203-202, MG, Brazil; deboragomide@epamig.br (D.R.G.);; 2Empresa de Pesquisa Agropecuária de Minas Gerais, Lavras 37203-202, MG, Brazil; 3Better Nature Research Center, Lavras 37203-016, MG, Brazil

**Keywords:** dairy cows, feed efficiency, grazing, reconstituted corn grain silage, particle size, starch

## Abstract

**Simple Summary:**

Rehydration and ensiling (reconstitution) allow for low-cost storage and can improve the nutritive value of the low-digestibility, high-vitreousness corn grain typical of Brazil. A large particle size increases the grinding rate but can reduce starch digestibility. This experiment evaluated the effect of mature flint corn finely or coarsely ground ensiled for 40 days or finely ground grain on the lactation performance, intake, ruminal fermentation, and digestibility of dairy cows fed on palisade grass pasture, whole plant corn silage, and concentrates. Fine grinding of ensiled corn grain increased the feed efficiency and coarse grinding reduced the total tract starch digestibility.

**Abstract:**

The particle size (PS) of reconstituted corn (REC) can affect the grinding rate and starch digestibility in dairy cows. We evaluated the effect of the PS of REC ensiled for 40 days on the pasture dry matter intake (DMI), lactation performance, total tract digestibility, and ruminal fermentation of grazing dairy cows. The treatments were coarse REC (CO, 1694 µm), fine REC (FI, 1364 µm), or finely ground (GC, 366 µm) flint corn (68% vitreousness) at 29.6 ± 1.4% of diet DM (mean ± SD). Eighteen dairy cows (mean milk yield 21.3 kg/d) were split into three groups by production level and were assigned within each group to a sequence of treatments in 3 × 3 Latin squares of 21-day periods. Cows were individually fed a constant amount of whole-plant corn silage 3 ×/d (2.7 kg DM/d) and corn treatments and soybean meal according to their group. There was no significant interaction between treatment and the production level. Cows fed FI had a lower DMI (16.7 vs. 18.1 kg/d) than those fed GC, and both did not differ from CO (17.7 kg/d). There was no treatment effect on milk yield (mean: 19.2 kg/d). Cows fed CO had the lowest total tract digestibility of starch (86.3 vs. 92.3% of intake) and the highest fecal starch concentration (7.0 vs. 4.0% of DM). The NDF digestibility was lower for GC-fed cows than CO- and FI-fed cows. Plasma glucose was higher in cows fed FI and CO (75.0 mg/dL) than those fed GC (70.8 mg/dL). Ruminal volatile fatty acids and the pH did not differ. Fine grinding of REC increased the feed efficiency relative to CO and GC. Coarse grinding of REC ensiled for 40 days reduced the total tract starch digestibility relative to FI and GC.

## 1. Introduction

Grazing dairy systems are considered environmentally and animal welfare friendly and can produce milk with a desirable composition for consumers; therefore, there is potential for market capitalization on dairy products from grazing cows [1]. However, despite the high potential milk yield per area of tropical pasture [2], the high concentration of neutral detergent fiber (NDF) in intensively managed tropical pastures [3] may limit the dry matter intake (DMI) and energy supply in cows fed exclusively on grass [4]. The supplementation of grazing dairy cows with energy concentrates can increase their DMI and the productivity per animal and per area, even with a decrease in pasture intake [5].

The response in the DMI, lactation performance, and rumen fermentation of grazing cows to the supplementation with high-energy cereal concentrates may be affected by the grain type and processing [6], the amount and frequency of supplementation [7,8], and the pasture availability [9,10]. Concentrates may be offered to grazing cows alone or mixed with a restricted amount of supplemental forage [11], such as whole plant corn silage (WPCS) [12,13] or hay [14]. The supplementation of grazing cows with concentrates mixed with forage may reduce the rate of passage of digesta through the digestive tract [14] and can reduce ruminal ammonia and increase the pH [6,15] compared to cows fed concentrates alone at a low frequency per day. Besides this, this practice allows grazing animals to have a daily feeding period in a heat abatement facility, which may be desirable during the hot–rainy season of tropical regions, without a high investment in facilities for total confinement.

The Brazilian corn seed industry prefers kernels with a hard texture endosperm (flint or vitreous), inherently of low starch digestibility in the rumen and in the total digestive tract [16,17]. Flint corn has a high concentration of prolamins in the endosperm, which are hydrophobic proteins that surround the starch granules and can reduce starch digestibility [18]. Rehydration and ensiling (reconstitution) of ground mature corn kernels (REC) is a low-cost method for grain storage in farms and can increase starch digestibility [19,20] by degradation of prolamins by microbial and plant proteases during ensiling [21,22,23]. Increased starch digestibility can improve the lactation performance and feed efficiency (milk/DMI) of dairy cows [24].

The duration of storage can affect the operational and economic efficiency of REC. Short storage allows for ensilage and fast use throughout the year, reducing the working capital requirement for grain purchase and storage, and enables quick use of farm-harvested grain. However, short storage may reduce the desirable positive effects of ensiling on prolamin degradation and starch digestibility [25]. Another determinant of the operational efficiency in the production of REC is the degree of grinding. A higher grinding rate is obtained when corn is coarsely ground, saving time, labor, and grinding energy [26]. However, coarse grinding can reduce starch digestibility by reducing the surface area for the action of amylases of microbial and endogenous origin in the digestive tract [27]. The beneficial effects of fine grinding and long storage on REC digestibility have been demonstrated in vitro [28,29,30,31,32].

However, when the REC of flint corn was stored for more than 205 days, the degree of grinding affected the total tract starch digestibility of dairy cows fed on total mixed rations only when the dietary starch concentration was high (29.2% of DM) but had no effect when the dietary starch was low (23.5% of DM) [26]. Batalha [33] also observed no effect of the particle size of REC stored for 177 days on the lactation performance, intake, and feed efficiency of low-producing dairy cows fed on tropical pasture. Nonetheless, to the best of our knowledge, the effect of REC particle size from flint corn stored for short durations on the performance of dairy cows fed tropical pasture has not been evaluated. As grain supplementation is provided at a low frequency per day for grazing dairy cows, finely ground REC may cause pulses of fermentation in the rumen, inducing ruminal acidosis and decreases in fiber digestion, pasture intake, and milk fat concentration. Conversely, greater ruminal starch fermentability may improve the feed efficiency and milk protein synthesis. Therefore, the objective of this experiment was to evaluate the effect of finely or coarsely ground REC ensiled for 40 days on the lactation performance, DMI, nutrient digestibility, and ruminal fermentation profile of grazing dairy cows. We hypothesized that for the REC of flint corn stored for short durations (40 days), fine grinding (3 mm mesh diameter screen) would increase starch digestibility and the feed efficiency relative to coarse grinding (8 mm mesh screen). Fine mature flint corn ground at a frequently used particle size (1.5 mm mesh screen) was also evaluated as a control treatment.

## 2. Materials and Methods

The experimental protocol was approved by the University of Lavras Bioethics Committee in Utilization of Animals (protocol number 051/17).

### 2.1. Farm Location and Environment

The experiment was conducted at the Experimental Field of Três Pontas at the Minas Gerais Agricultural Research Center (EPAMIG), Três Pontas, MG, Brazil (21°20′25.92″ S, 45°28′46.01″ W), from 15 January to 18 March 2018. The farm is located at 905 m above sea level and has a humid subtropical climate Cwa according to the Köppen–Geiger classification. The temperature and humidity were measured at hourly intervals during the experiment (1483 recordings; Appendix A). The temperature was 22.3 ± 3.6 °C (14.5 to 31.4) (mean ± SD (minimum to maximum)) and the humidity was 76.0 ± 7.8% (31 to 96). The temperature–humidity index (THI) was calculated according to Youlsef [34] as: T + 0.36 × DP + 41.2, where T = the temperature (°C) and DP = the dew point (°C). The THI was 69.8 ± 3.6 (60.4 to 78.0). The daily precipitation and mean THI throughout the experiment are reported in Figure 1.

### 2.2. Pasture Management and Evaluation

Two hectares of pasture of *Urochloa brizantha* cv. Marandu (palisade grass) was managed under a rotational grazing system. The area was divided into 21 paddocks of 910 ± 98.8 m^2^ and each paddock was grazed for one day at 20-day intervals. All paddocks had access to a common area with water troughs. After each grazing, the paddocks were mowed at 30 cm height and manually fertilized with 15 kg of NPK (20:05:20) to achieve approximately 32 kg of N/ha per 21-day grazing cycle throughout the experiment. The same seven paddocks were grazed during the seven days of data collection in each of the three experimental periods.

On the first day of each experimental period, five sampling squares (0.70 m^2^) were selected to represent different heights of pasture within the seven paddocks grazed during the last week of each period. The forage height in each sampling square was measured with a circular acrylic ascending plate of 32 cm diameter and 350 g, and forage was harvested at ground level. Samples were dried in a forced air oven at 55 °C for 72 h for DM determination. A linear regression was generated based on the 15 samples obtained during the experiment for the prediction of dry pasture mass based on disk meter height: ton of DM/ha = 5.3866 + 0.1919 × height (cm) (Figure 2). The forage height in 50 sampling spots was evaluated in each paddock pre-grazing, post-grazing, and post-mowing throughout the experiment, and the forage mass was estimated daily [35]. The pre-grazing and post-grazing pasture masses are reported in Figure 3. The stocking and volumetric density, forage mass, pasture allowance and intake, and grazing efficiency were calculated for the seven paddocks grazed during the week of data collection in each experimental period [36].

### 2.3. Corn Grain Processing

The REC was ensiled nine times with seven-day intervals in 200 L plastic buckets to achieve the 40 ± 3 days duration of storage throughout the experiment. Kernels from the same batch of a mature corn hybrid were ground with a stationary hammer mill (Nogueira TN-8, Nogueira Máquinas Agrícolas, São João da Boa Vista, SP, Brazil) using diameter mesh screens of 1.5 mm for finely ground mature corn (GC), 3 mm for finely ground REC (FI), and 8 mm for coarsely ground REC (CO). On each ensiling day, a sample of 100 g kernels was obtained, and a composite sample was formed for the entire experiment. From the composite sample, 100 kernels were separated to form five groups of kernels visually homogenous in morphology. Then, a sample of 10 kernels was selected in proportion to the weight of each group for dissection of the endosperm. Kernels were immersed in distilled water for five minutes and dried with a paper towel. The germen, pericarp, and endosperm were manually separated using a scalpel and dried at 105 °C for 12 h. The floury endosperm was dissected with a scalpel and kernel vitreousness was defined as the proportion of vitreous endosperm in the total endosperm [16]. The vitreousness of ten kernels was 68 ± 5% of endosperm, and the proportions of the endosperm, germen, and pericarp were 86.5 ± 1.2, 9.6 ± 0.6, and 3.9 ± 1.4% of the kernel DM, respectively.

For each 200 L bucket, ground corn (148 kg) was mixed in a 1.2 m^3^ vertical feed mixer with water (52 kg) and 3 g of silage inoculum (Kerasil+. *Lactobacillus plantarum* 2.6 × 10^10^ CFU/g, *Pediococcus acidilactici* 2.6 × 10^10^ CFU/g, *Propionibacterium acidipropionici* 1.4 × 10^10^ CFU/g; Kera Nutrição Animal, Bento Gonçalves, RS, Brazil) to achieve a 35% moisture concentration. Two 200 L buckets were filled per REC treatment per day and the weight of hydrated corn was 198.3 ± 1.1 kg/bucket (*n* = 36). Ensiled corn was enveloped in oxygen barrier plastic (Silostop, Lallemand Animal Nutrition, Aparecida de Goiânia, GO, Brazil), and the buckets were capped. The GC was ground and sampled weekly during the period of data collection and samples of ensiled corn from each bucket before and after ensiling were stored frozen. The DM concentration per bucket was measured by drying at 55 °C for 72 h and subsequently at 105 °C for 24 h. Thawed REC samples (25 g) were blended with distilled water (225 mL) for pH determination (HI 2210 pH meter, Hanna Brasil, Barueri, SP, Brazil).

The particle size distribution of each silo before and after ensiling (*n* = 18/REC) and for GC (*n* = 9) was evaluated. Samples were dried in a forced-air oven at 55 °C for 72 h before being dry sieved (127 ± 6.8 g) for 10 min with a Bertel shaker (Bertel Indústria Metalúrgica Ltd.a., Caieiras, SP, Brazil) with sieves of square diagonal apertures of 4750, 2830, 1400, 1000, 600, and 250 µm and a pan. The proportion of particles retained on each sieve was determined, and the geometrical mean particle size (GMPS, µm), particles per gram, and surface area (cm^2^/g) were calculated [37].

### 2.4. Lactation Performance Experiment

#### 2.4.1. Cows, Experimental Desing, and Feeding Management

Fifteen Holstein cows (six primiparous) and three rumen-cannulated 50:50 Holstein–Gyr cows formed six squares of three cows by breed, lactation order (1 vs. >1), and mean milk yield during the five days immediately before experiment initiation. The six squares formed three groups of two squares each based on milk yield: high (26.1 ± 2.6 kg/d milk yield; 83 ± 35 days in lactation; 603 ± 53 kg BW), medium (20.9 ± 2.0 kg/d milk yield; 257 ± 92 days in lactation; 563 ± 48 kg BW), and low (16.9 ± 1.0 kg/d milk yield; 230 ± 97 days in lactation; 522 ± 63 kg BW). Both medium and low groups had a square of primiparous cows and the high group had a square of rumen-cannulated cows. Within a square, cows were randomly allocated to a sequence of the three treatments (GC, FI, and CO) in 3 × 3 Latin squares, with 21-day periods, 14-day adaptation, and seven days of data collection.

Cows were milked 2 ×/d at 06.00 and 16.00 h and were individually fed with corn treatments, soybean meal, and a mineral–vitamin premix in sand-bedded tie stalls at 05.00, 10.00, and 14.00 h without feed refusals. Concentrated ingredients were manually mixed at the feed bunk for each cow and the amount offered was 1/3 of the diet calculated daily allowance per feeding. The WPCS (1.4 kg of DM/feeding) was mixed with concentrates only at the 10.00 h and 14.00 h feedings.

Cows grazed on a new paddock each day. The distance from the 21 paddocks to the milking parlor was 348 ± 87 m, ranging from 249 to 562 m. The average daily grazing time allowance was 15 h and 9 min (± 26 min). Cows went from the pasture to the milking parlor at 05.00 h and were sent back to the same paddock at approximately 07.00 h, after finishing the first daily concentrate feeding. At 10.00 h, cows were fed the second feeding with concentrates plus WPCS and were kept from approximately 11.00 to 14.00 h in a shaded resting area until the third daily feeding with the same feeds. Cows grazed on a new paddock after the afternoon milking.

#### 2.4.2. Treatment Diets

Treatment diets were formulated immediately before the experiment for each of the three production groups of six cows and fixed amounts of WPCS and concentrates were fed per cow throughout the experiment. The DMI was estimated for the average cow of each group (milk yield, BW, days in lactation, 3.8% milk fat) based on the NRC 2001 dairy model [38]: 20.4 kg/d for high, 18.6 kg/d for medium, and 16.6 kg/d for low. Diet formulation assumed a fixed amount of 2.8 kg DM/d from WPCS (35% DM on an as-fed basis and 8.8% CP and 45% NDF on a DM basis based on NIRS analysis; 3rLab/Rock River, Lavras, MG, Brazil). The pasture was assumed to contain 14% CP and 63% NDF on a DM basis. The intake of forage NDF (pasture + WPCS) was set at 1.1% of BW in all treatments. The CP concentration was formulated to be 16.1% in high, 16.0% in medium, and 15.4% in low production groups by using soybean meal and each corn treatment as concentrates. Equal amounts (kg/d) of mineral and vitamin sources were added to each diet. The predicted forage to concentrate ratios were 55:45 in high, 57:43 in medium, and 60:40 in low and the concentrations of corn treatments were 29.8, 28.4, and 27.0% of DM, respectively. The predicted amounts of corn DM were (kg/d) 6.1 on high, 5.3 on medium, and 4.5 on low.

#### 2.4.3. Intake and Diet Composition

During the third week of each experimental period, samples of WPCS and concentrate ingredients were obtained daily and frozen. A feed composite sample was formed per period. Samples of the pasture were obtained with a rumen-cannulated cow and manually by simulated grazing (hand-pluck method). On days 17 to 20 of each period, an additional rumen-cannulated cow had the rumen evacuated after the afternoon milking and was introduced into the newly grazed paddock with the other experimental cows for 50 min. The rumen digesta was collected and frozen and a composite sample was formed per period for analysis of ash, ether extract (EE), starch, NDF, and undigested NDF (uNDF). Pasture samples were simultaneously collected by simulated grazing for analysis of DM and crude protein (CP). The fecal output was estimated using Cr_2_O_3_ as an external marker (99.8% purity, Óxido de Cromo III Verde, Dinâmica Química Contemporânea, Indaiatuba, SP, Brazil) dosed at 9 g/cow/d. The marker was mixed with the concentrates offered to each cow at 05.00, 10.00, and 14.00 h (3 g/feeding) from day 8 to 21 of each period. Fecal sampling was performed on days 17 to 21 of each period at the time of concentrate supplementation (3 ×/d) and a composite frozen sample was formed per period. Fecal and marker Cr concentrations were analyzed by atomic absorption spectroscopy [39]. The uNDF concentration (% of DM) in feces and feed ingredients, including the pasture harvested by simulated grazing and by a rumen-cannulated cow, was evaluated by in situ incubation for 240 h [40]. Pasture DMI (kg/d) was estimated with the uNDF concentration of the sample harvested by a rumen-canulated cow as
Pasture DMI (kg/d) = [fecal uNDF excretion − (WPCS uNDF intake + corn grain uNDF intake + soybean meal uNDF intake)]/pasture uNDF concentration.(1)

Samples of feeds and feces were dried in a forced-air oven at 55 °C for 72 h and ground to pass in a 1 mm diameter mesh screen (Wiley mill, Thomas Scientific, Swedesboro, NJ, USA). The DM concentration was determined by drying at 100 °C for 24 h and the ash concentration was determined by incineration at 550 °C for 8 h. The CP concentration was determined with a micro Kjeldahl steamer distiller [41], the ash-free NDF concentration was determined by filtration in porous crucibles with heat-stable α-amylase and sodium sulfite [41], and EE was determined [41]. Starch and free glucose were analyzed with α-amylase and amyloglucosidase and colorimetry was used for glucose [42,43]. Data from feed and fecal analyses were used to calculate the concentration of non-fiber carbohydrates (NFC): 100 − (CP + NDF + EE + ash). The composition of the diets in nutrients and ingredients was the intake of all cows on a treatment divided by the total DMI of each treatment.

#### 2.4.4. Milk Yield and Composition

From day 18 to 21 of each experimental period, milk was measured and samples were collected in proportion to the amount produced in each milking. Samples were stored in flasks containing 2-bromo-2-nitropropane-1-3 diol and refrigerated until shipment to a commercial laboratory (Laboratory of the Paraná State Holstein Breeders Association, Curitiba, PR, Brazil). Milk CP, fat, lactose, total solids, somatic cell count (SCC), and milk urea-N (MUN) were analyzed by mid-infrared analysis (NexGen FTS/FCM; Bentley Instruments, Chaska, MN, USA). Milk energy secretion (Mcal/d) was calculated as [44]: [(0.0929 × % fat) + (0.055 × % crude protein) + (0.0395 × % lactose)] × kg of milk. The secretion of energy-corrected milk (ECM; kg/d) was calculated as Milk energy secretion/0.70 (assumes 0.70 Mcal/kg of milk with 3.7% fat, 3.2% crude protein, and 4.6% lactose). The 4% FCM (FCM; kg/d) was calculated as [45]: 0.4 × kg of milk + 15 × kg of fat.

#### 2.4.5. Fecal Viscosity, Digestibility, and Feed Efficiency

Fecal viscosity was measured according to an adapted protocol from Cannon et al. [46]. Briefly, fecal samples (100 g) were diluted with 130 mL of distilled water, homogenized for 30 s, and strained through two layers of cheesecloth. The viscosity was measured in three 60 mL aliquots of the filtered solution with a Brookfield Viscometer at 60 rpm (Brookfield Ametek, Middleboro, MA, USA). The total tract apparent digestibility of DM, OM, NDF, and starch was estimated from the calculated intake and fecal excretion of nutrients, as previously described. The digestible OM intake (DOMI) and feed efficiencies were calculated: Milk/DMI, ECM/DMI, and ECM/DOMI.

#### 2.4.6. Ruminal Fermentation

Although the experimental power to detect statistically significant differences was limited by cow availability, ruminal fluid was obtained from the three rumen-cannulated cows from day 18 to 21 of each period at 05.00, 10.00, and 14.00 h (before each concentrate feeding). The pH was immediately measured (pHmetro Digimed DM20; Datamed Instrumentos Científicos e Médicos, Belo Horizonte, MG, Brazil). Samples of strained ruminal fluid were frozen in liquid nitrogen to stop fermentation and stored at −20.0 °C After thawing and centrifuging at 4 °C and 8855× *g* for 15 min, composite samples were pooled per cow, sampling time, and period for analysis for volatile fatty acids (VFA) and ammonia-N. Samples were analyzed for VFA by gas–liquid chromatography (CP 3800 Gas Chromatography Varian; Varian Chromatography Systems, Palo Alto, CA, USA) with a capillary column (CP-Wax 58 (FFAP) CB; Varian Analytical Instruments, Palo Alto, CA, USA). Ammonia-N (mg/dL) was analyzed with a colorimetric assay catalyzed by indophenol [47].

#### 2.4.7. Blood Samples

Blood samples from the coccygeal vessels were obtained from day 19 to 21 of each period at 05.00, 10.00, and 14.00 h (simultaneous with rumen sampling). Samples were obtained in vacutainers containing EDTA for examining plasma urea-N (PUN) and in vacutainers with potassium fluoride for examining glucose. Blood samples were centrifuged at 2000× *g* for 10 min at room temperature. Plasma was obtained and frozen at −20 °C. After thawing, composite samples were formed per cow, sampling time, and period for the determination of urea (Uréia Enzimática K047, Bioclin, Belo Horizonte, MG, Brazil) and glucose (Glicose Enzimática Líquida, Doles Reagentes e Equipamentos para Laboratórios, Goiania, GO, Brazil). The PUN concentration was obtained by multiplying the urea value by 0.466.

### 2.5. Ruminal In Situ Degradation

The ruminal in situ degradation of corn DM was evaluated in nine samples of GC (1/week) and 72 samples of REC (2 silos/week) before and after ensiling (36/REC). Samples were incubated in two rumen-cannulated cows in mid-lactation (23.4 ± 2.6 kg/d of milk) and fed a total mixed ration based on corn silage, soybean meal, and ground corn. Samples were dried at 55 °C for 72 h and 5.32 ± 0.24 g was inserted into 10 × 20 cm non-woven textile bags (pore size 100 µm; 100 g/m^2^). The incubation times were 0, 3, 6, 18, and 48 h (duplicate bags for each time point/cow). Time 0 bags were washed for 30 min in tap water at room temperature and were immediately frozen. Bags were soaked for 30 s in warm water before ruminal incubations. The incubated bags were immersed in ice-cold water and immediately frozen after removal from the rumen. All bags were unfrozen at the same time and washed in two washing machine cycles. A two-pool model was used to describe corn DM degradation kinetics [48]. The rapidly degradable A fraction (% of DM) was the 0 h bag disappearance and the slowly degradable B fraction was 100 − A. The fractional degradation rate of fraction B (kd, %/h) was the slope of the linear regression of the natural logarithm of the bag residue as a proportion of the incubated sample size from time 0 to 48 h. The effective ruminal degradation (ERD) was: A + B × [kd/(kd + kp)], where kp is the fractional passage rate of concentrates (6.5%/h) estimated for the average experimental cow and diet [49]. The mean value of the four incubated bags per incubation time per treatment (two/cow) was used in statistical analyses.

### 2.6. Statistical Analysis

Data were analyzed with the MIXED procedure of SAS (University Edition. SAS Institute Inc., Cary, NC, USA). Silage variables (particle size, ruminal in situ degradation, pH, and DM concentration) were analyzed as a completely randomized design with a model containing the fixed effect of treatment (GC, fine before ensiling, coarse before ensiling, fine after ensiling, and coarse after ensiling).

The Latin square model had the random effects of group (high, medium, and low) and cow nested within group (1 to 18), the fixed effects of period (1 to 3) and treatment (GC, FI, and CO), and the interaction between treatment and group. For variables obtained over time (PUN and glucose) the fixed effect of time (05.00 h, 10.00 h, and 14.00 h) and the two-term and three-term interactions between time, treatment, and group were added to the previous model. For the rumen variables evaluated over time in one Latin square (pH, VFA, and ammonia-N), the model contained the random effect of the cow (1 to 3) and the fixed effects of period (1 to 3), treatment (GC, FI, and CO), and time (05.00 h, 10.00 h, and 14.00 h), and the interaction between treatment and time. The whole-plot error for the repeated measure variables was the interaction between cow, period, and treatment. For each variable, the best covariance structure was defined by Schwarz’s Bayesian criteria among first-order autoregressive, compound symmetry, and unstructured. Degrees of freedom were calculated with the Kenward–Roger option. Significance was declared at *p* ≤ 0.05 and tendencies at 0.05 < *p* ≤ 0.10. Treatment means were compared with the Tukey–Kramer test.

## 3. Results

### 3.1. Environment and Pasture Variables

A THI greater than 68, suggestive of the occurrence of heat stress [50], was observed in at least one hour of the day on 62 of the 63 days of the experiment. The frequency of a THI ≥ 68 was 69% of the time and a THI ≥ 72 occurred 65% of the time. Cows grazed in a hot environment, typical of the hot–rainy season of southeast Brazil. Precipitation was more intense during data collection in the second experimental period than in periods 1 and 3 (Figure 1).

Table 1 presents data on each paddock obtained with the ascending plate during the seven days of data collection on each experimental period. The daily forage allowance per cow (563 kg BW) was 54.8 kg of DM at 11.1 ± 1.1 cows/ha and 43.3 m^2^/d/cow (Table 1). *Urochloa brizantha* cv. Marandu (palisade grass) was grazed at 37.7 cm pre-grazing height, equivalent to 12,622 kg of DM/ha of forage allowance, and 30.1 cm post-grazing height. The estimated daily pasture intake based on pre-grazing and post-grazing pasture heights was 7.5 kg of DM/cow. There was a numerical trend for the daily pasture allowance (kg of DM/ha) to be reduced with the increase in experiment duration when cows grazed from mid-January to mid-March, and the highest allowance during the first grazing period reflects the lack of pasture mowing before the start of the experiment (Figure 3).

The pasture composition is reported in Table 2. The CP concentration evaluated by simulated grazing (hand-plucking) was 12.7% of DM and the NDF analyzed in samples obtained with a rumen-cannulated cow was 56.8% of DM, both lower than the values used for pre-experimental ration formulation (14% and 63%, respectively). The NDF concentration of samples obtained by simulated grazing was 53.8% of DM. The WPCS also had a lower CP (6.4 vs. 8.8% of DM) and NDF (42.2 vs. 45% of DM) than the sample analyzed before the experiment by NIRS. The DM concentration of REC was close to the predicted concentration, although slightly lower (63.1 for FI and 63.9 for CO vs. 65% of as-fed). The uNDF concentration of the sample harvested with a rumen-cannulated cow was 13.5% of DM (Table 2) and it was 11.8% of DM for the sample obtained by simulated grazing. The correlation of uNDF concentration measured by the two sampling procedures was high (r^2^ = 0.99) and uNDF was consistently higher in samples harvested with a cow than by simulated grazing (Appendix A). The uNDF concentration was higher during period 1 than in periods 2 and 3, denoting the positive relationship between pasture allowance and uNDF concentration.

### 3.2. Pasture Intake and Diet Composition

The pasture intake predicted in the sample collected by simulated grazing was higher than the prediction based on samples harvested with a rumen-cannulated cow (Appendix A). Data obtained from the cow were used to estimate pasture intake (Table 3) and the composition of the consumed treatment diets (Table 4). The estimates of pasture intake obtained by both sampling procedures and marker data per cow were close to the mean pasture intake estimated from the pre-grazing pasture height subtracted from post-grazing pasture height (Table 1). There were no treatment or production group effects on pasture intake (mean of 7 kg of DM/d, *p* ≥ 0.16). The dietary pasture concentration was 39.6 ± 2.7% of DM for the nine consumed diets calculated after pasture intake was estimated (Table 4). The concentration in the diet of total forage (WPCS plus pasture) were 55.2 ± 2.4% of DM and the corn grain concentration was 29.6 ± 1.4% of DM. Diet CP was 14.8 ± 0.5, NDF was 33.2 ± 1.1, forage NDF was 29.0 ± 1.3, and starch was 22.6 ± 1.1% of DM.

### 3.3. Particle Size and Ruminal Degradation of Silages

The GC kernels had a lower GMPS and more particles/g and surface area than FI and CO before and after ensiling (Table 5). The GMPS of GC was 366 µm, FI before ensiling was 1364 µm, and CO before ensiling was 1694 µm. Ensiling reduced the GMPS of FI (1258 µm) and did not affect the GMPS of CO (1648 µm).

The ruminal in situ degradation of corn DM was affected by treatment (Table 5). When a constant kp was assumed for all treatments (6.5%/h), FI and CO before ensiling had a lower ERD than FI and CO after ensiling and no difference was detected between silages differing in particle size. The GC had a higher ERD than CO before ensiling (*p* ≤ 0,05) and there was a tendency to have a higher ERD than FI before ensiling (*p* ≤ 0.10), but this did not differ from the silages independently of silage particle size. Coarsely ground flint corn (8 mm screen) was less degradable in the rumen than finely ground corn (1.5 mm screen) before ensiling, but ensiling eliminated the effect of particle size on corn ERD.

Ensiling increased the silage kd and particle size did not affect the kd either before or after ensiling (Table 5). The kd of GC was intermediate and did not differ from the other four treatments. The pool size of fraction A was higher in GC and ensiled corn than in corn before ensiling, independently of the particle size of corn ground for silage. The 3 h ruminal incubation was not effective in detecting differences in ruminal in situ degradation of ground corn kernels differing in processing and particle size. The silage pH and DM concentration did not differ by treatment.

### 3.4. Intake and Lactation Performance

Intake, lactation performance, and feed efficiency are reported in Table 3. There was no interaction between treatment and production group for the variables evaluated (*p* ≥ 0.28). Cows fed with FI had a lower DMI and DOMI than cows fed with GC (*p* ≤ 0.05), and cows fed with CO were intermediate and did not differ from the other treatments independently of their production group. However, the yields of milk and components did not differ by treatment diet (*p* ≥ 0.50). Cows fed with FI had a higher feed efficiency (milk/DMI and ECM/DMI) than cows fed with GC and CO (*p* ≤ 0.05). When expressed as per unit of energy intake estimated by DOMI, cows fed with FI were more efficient than cows fed with GC (*p* ≤ 0.05). Fine grinding of REC in a short duration of ensiling increased the feed efficiency of grazing dairy cows by reducing the DMI at similar milk yields. As expected, cows in the high group had a higher lactation performance, DMI, and feed efficiency than cows grouped as low (*p* ≤ 0.05).

### 3.5. Total Tract Digestibility, Blood Glucose, and Nitrogen Metabolism

No interaction was detected between treatment and group (*p* ≥ 0.13) for diet digestibility, plasma glucose, and PUN (Table 6). The total tract apparent digestibility of DM, OM, and starch was lower in cows fed with CO than in cows fed with GC and FI (*p* ≤ 0.05). A similar trend was detected for the fecal starch concentration; it was higher in cows fed with CO than in cows fed with GC and FI (*p* ≤ 0.05). The NDF digestibility was higher (*p* ≤ 0.05) in cows fed with FI (48.9% of intake) than GC (45.6% of intake) and there was a tendency (*p* ≤ 0.10) to be higher in FI than CO (45.9% of intake). Cows grouped as high had a lower starch digestibility (87.8% of intake) and a higher concentration of fecal starch (6.4% of DM) than medium and low groups (91.6% of intake and 4.3% of DM, respectively; *p* ≤ 0.05), but no effect of group was detected for the digestibility of DM, OM, and NDF (*p* ≥ 0.34). Fecal viscosity did not differ by treatment nor group (*p* ≥ 0.27).

Cows fed with GC had lower plasma glucose concentration than cows fed with FI and CO (*p* ≤ 0.05) and no effect of production level was detected for blood glucose (*p* = 0.49). There was no effect of treatment or group on PUN (*p* ≥ 0.71, Table 6), in agreement with the lack of a treatment effect on MUN (*p* = 0.44, Table 3) and ruminal ammonia-N (*p* = 0.52, Table 7).

### 3.6. Ruminal Fermentation

Data on ruminal VFA concentration and molar proportions obtained in one square of rumen-cannulated cows are presented in Table 7. There was no interaction between treatment and time (*p* ≥ 0.51) for rumen fluid sampled 3 ×/d at the same time as concentrate feeding at 05.00, 10.00, and 14.00 h. The rumen VFA concentration and profile, the acetate to propionate ratio, and pH did not differ by treatment diet (*p* ≥ 0.22).

The sampling time affected (*p* ≤ 0.01) the molar proportions of acetate, propionate, and isobutyrate, the acetate to propionate ratio, and the ruminal pH (Table 7). Acetate, isobutyrate, the acetate to propionate ratio, and pH were reduced over time, and propionate increased over time (*p* ≤ 0.04 for the comparison between all sampling time means). Values at 05.00, 10.00, and 14.00 h were 62.6, 61.6, and 60.1% of the total molar VFA, respectively, for acetate; 23.2, 24.5, and 26.0%, respectively, for propionate; and 1.25, 1.06, and 0.88%, respectively, for isobutyrate. The acetate to propionate ratios were 2.80, 2.66, and 2.45, respectively. The ruminal pHs were 6.4, 6.1, and 5.8, respectively. The ruminal pH was the lowest and the acetate to propionate ratio was the highest in samples obtained at 05.00 h when the cows came from night grazing, and the opposite was observed at 1400 h when cows were fed the third concentrate feeding.

## 4. Discussion

The animal stocking rate per ha of *Urochloa brizantha* cv. Marandu was 11.1 cows or 13.9 animal units (450 kg) in our experiment, similar to the suggested potential stocking rate of 15 animal units/ha [51]. The pre-grazing (37.7 cm) and post-grazing (30.1 cm) pasture height were higher than the optimum recommendation for Marandu palisadegrass based on the leaf area index [52,53]. The pasture intake measured with markers was 7 kg of DM/cow/d at a mean forage allowance of 55 kg of DM/cow/d, resulting in a 12% grazing efficiency, considered low for Marandu palisadegrass and in agreement with the high pasture allowance per cow [54]. The pasture allowance was 7.86 times the pasture intake. The pasture quality may have been negatively affected by the high allowance, since the uNDF concentration in pasture DM was positively correlated to the pasture allowance across all experimental periods. As a result of the low pasture utilization and the deterioration in pasture quality at a high pasture allowance, a practical recommendation was proposed to provide a pasture allowance of two times the expected pasture DMI or 25 kg DM/cow/d of pasture allowance when cows are fed with supplement [6]. Corn supplementation was evaluated at a high pasture allowance in our experiment; the effect of the supplement on grazing behavior, substitution rate, and DMI was not affected by restricted pasture availability [9,10].

The pasture CP was 12.7% of DM and the NDF was 56.8% of DM during the seven days of data collection in each experimental period. The mean concentrations observed for 19 samples of intensively managed Marandu palisadegrass obtained by simulated grazing was 13.9% CP and 60.7% NDF in DM [3], close to our values, suggesting that pasture composition was not seriously negatively affected by the high forage allowance in our experiment. The Holstein cows performed reasonably well in the hot environment of the experiment when compared with data from a commercial dairy farm milking Holstein cows fed on rotational grazing of tropical grass and supplemented according to requirement with concentrate feedstuffs [55]. The mean milk yield was 19.2 kg/d at 221 days in lactation, the mid-point of the 63-day experiment. Concentrate and forage supplementation occurred during the warmer hours of the day, and this seems to be a reasonable strategy for Holstein cows grazing in a tropical environment. Grazing occurred during short periods in the morning, late afternoon, and night.

The pasture intake in this farm management scenario was lower than predicted during ration formulation (7 vs. 7.8 kg of DM/d). The pasture intake based on plant height and the uNDF concentration of samples obtained by simulated grazing or with a rumen-canulated cow were similar. The two sampling procedures for the evaluation of uNDF were highly correlated and suggestive that samples obtained by hand-plucking had a higher nutritive quality than samples harvested with a rumen-cannulated cow, as judged by the lower NDF and uNDF concentrations of the former. The pasture NDF concentration was lower than predicted during ration formulation (56.8 vs. 63.0% of DM), as was the NDF concentration of WPCS (42.2 vs. 45.0% of DM). The average dietary concentration of forage NDF was lower than predicted (29.0 vs. 32.5% of DM), contributing to a lower than predicted actual intake of forage NDF (0.90 vs. 1.10% of BW). The intake was apparently not limited by excessive filling of the digestive tract by forage NDF and, in theory, the DMI was dictated by the fermentability and energy content of the supplement [56,57].

Based on the in situ estimates, the ERD of ensiled corn (FI and CO) was equivalent to GC when a constant kp (6.5%/h) was assumed for all corn sources. The pool size of fraction A and the kd of fraction B did not differ between treatments (GC, FI, and CO). Ensiling for 40 d increased the ERD, fraction A, and the kd of flint corn, with a GMPS greater than 1364 µm. Considering that finely ground mature corn may have a faster passage rate than the silage of high-moisture coarsely ground corn [58], the actual ERD may have been lower in GC than in FI and CO. A lower ruminal starch degradation could potentially reduce the ruminal outflow of propionate, a plausible explanation for the observed lowest plasma glucose concentration and highest DMI in GC. Propionate from ruminal starch degradation is a known glucogenic precursor and depressor of DMI [59]. An increased passage of finely ground, ruminal undegraded starch for GC may have been compensated for by an increased proportion of intestinal starch digestion [58], and the total tract starch digestibility was not negatively affected for GC relative to FI.

However, the total tract starch digestibility was lower for CO than for GC and FI, suggesting that the larger particle size of CO negatively affected the total tract starch digestibility relative to FI. The large particle size of CO may have decreased the intestinal starch digestibility [60], apparently with no major effect on ruminal starch degradation, as judged by the in situ ERD and the rumen fermentation variables. The ruminal VFA profile and pH were not affected by corn processing in our experiment, in agreement with studies evaluating different particle sizes of REC using longer storage durations [26,33]. The ruminal fermentation profile has not been shown to be very responsive to variations in the type of concentrate supplementation in grazing dairy cows [6]. The concentrations of ruminal ammonia-N, MUN, and PUN also did not differ by treatment. A reduction in ruminal ammonia-N was found to be the most consistent outcome when more rumen fermentable starch was fed to grazing dairy cows [6]. The plasma glucose of cows fed with FI and CO was higher than those fed with GC, although only FI induced a decrease in the DMI relative to GC. The fecal starch concentration was the highest and the total tract starch digestibility was the lowest in the high group, suggesting that an increased passage rate of starch may have reduced starch digestibility in cows with a high DMI [61]. The corn processing method and the cow production level (group) induced changes in starch digestion.

Milk components (fat, protein, and lactose) did not differ by treatment, but a low milk fat percentage occurred in all treatments (2.99% of milk), suggesting that ruminal acidity was a common feature of the diets evaluated. The low daily frequency of concentrate feeding, even at 23% starch in diet DM (not excessively high), may have contributed to the relatively low pH observed in all treatments (6.08 on average). The rumen pH was lowest immediately before the third concentrate feeding of the day (5.8) and highest immediately before the first supplementation in the morning (6.4) when cows came from the night grazing period. The use of feed additives to control sub-acute ruminal acidosis may deserve evaluation as an alternative for grazing dairy cows fed starchy concentrates at a low daily frequency.

Cows fed GC had a higher DMI and a similar milk yield than cows fed FI. The pasture intake was numerically lower for FI compared to GC, although the difference was not statistically significant, suggestive of a greater substitution rate of pasture by concentrates on cows fed FI, since concentrates and WPCS were fed in a restricted amount and were totally consumed during the experiment. Although rumen fermentation variables do not support a significant difference between FI and GC in ruminal starch degradation, more rumen fermentable starch is a plausible explanation for the lower DMI in cows fed FI than in cows fed GC [59]. Curiously, the total tract NDF digestibility was higher for cows fed FI than those fed GC, with no difference in ruminal pH. It has been postulated that an increased ruminal availability to fiber-digesting bacteria of starch hydrolysis products could improve NDF digestibility [62]. Cows fed FI had a higher feed efficiency than cows fed GC, driven by a reduced DMI at a similar milk yield. The energetic efficiency (ECM/DOMI) was also increased by FI relative to GC. When lactation performance and feed efficiency are considered, the coarse grinding of REC stored for 40 days was adequate for the supplementation of grazing dairy cows, although the total tract starch digestibility was negatively affected. The feeding of coarsely ground REC stored for short durations to high-producing dairy cows fed on total mixed rations needs further study.

## 5. Conclusions

There was no detectable relationship between corn processing and cow production level. There were no treatment effects on the ruminal fermentation profile, pH or the secretion of milk components. The coarse grinding of REC reduced the total tract starch digestibility relative to GC and FI, but did not affect the DMI, milk yield, or feed efficiency. The fine grinding of REC ensiled for 40 days increased the feed efficiency (milk/DMI) relative to CO and GC.

## Figures and Tables

**Figure 1 animals-13-01932-f001:**
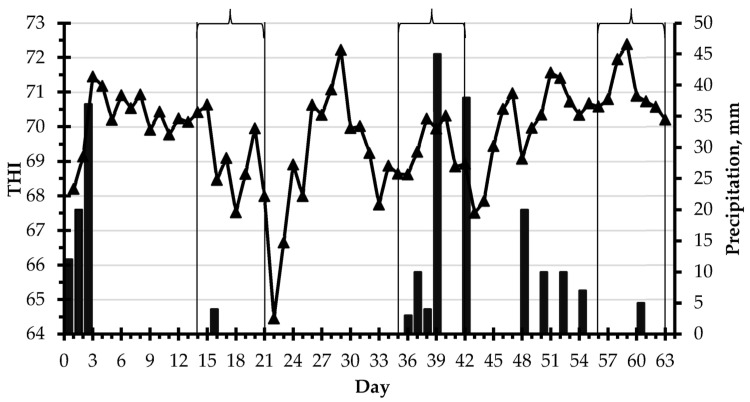
Daily mean temperature–humidity index (THI ▲) and precipitation (■) during the experiment. Days of data collection on each experimental period are shown in brackets.

**Figure 2 animals-13-01932-f002:**
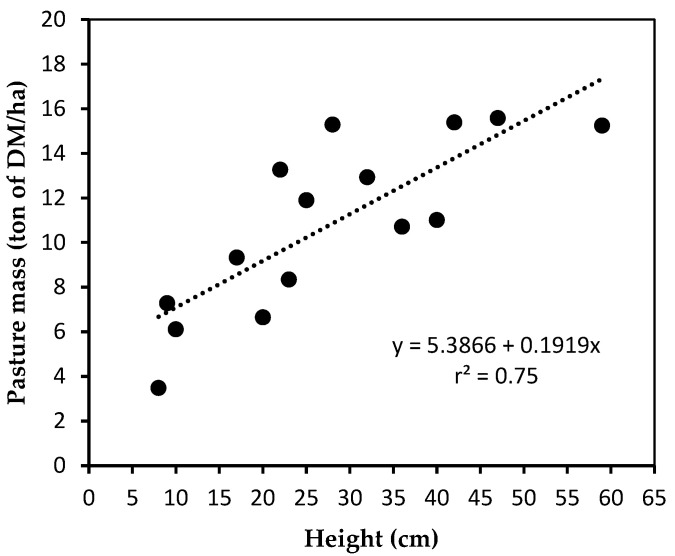
Linear regression for prediction of dry pasture mass (ton of DM/ha) based on disk meter height (cm).

**Figure 3 animals-13-01932-f003:**
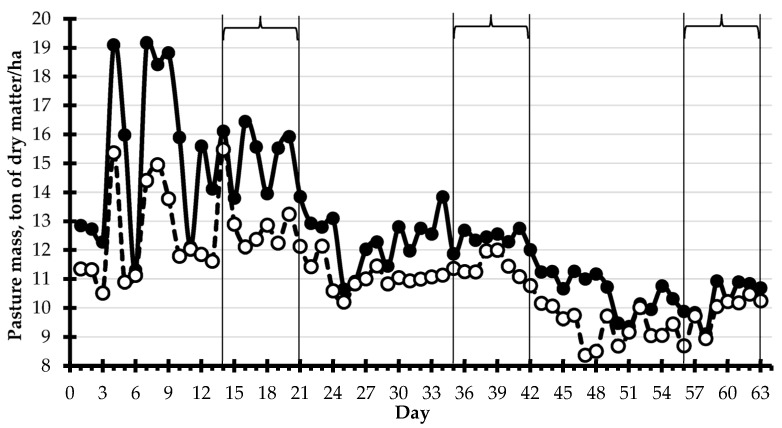
Pasture mass pre-grazing (
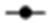
) and post-grazing (
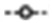
) during the experiment. Days of data collection in each experimental period are shown in brackets.

**Table 1 animals-13-01932-t001:** Mean and standard deviation of each paddock (1 to 7) of *Urochloa brizantha* cv. Marandu grazed for one day during the last week of each experimental period on day 21 of regrowth (*n* = 3).

Paddock	1	2	3	4	5	6	7	Mean ± SD
Paddock size, m^2^	853	968	990	920	874	879	819	910 ± 97.2
Available area ^1^, m^2^/d								
Per cow (563 kg)	40.6	46.1	47.1	43.8	41.6	41.9	39.0	43.3 ± 4.6
Per 450 kg BW	32.5	36.8	37.7	35.0	33.3	33.5	31.2	34.6 ± 3.7
Per 100 kg BW	7.2	8.2	8.4	7.8	7.4	7.4	6.9	7.7 ± 0.8
Stocking density, animals/ha/d								
Cows (563 kg)	246.2	216.9	212.1	228.3	240.3	238.9	256.4	233.1 ± 22.6
450 kg BW	308.0	271.4	265.4	285.6	300.6	298.9	320.8	291.6 ± 28.3
100 kg BW	1386.0	1221.4	1194.2	1285.1	1352.7	1345.1	1443.6	1312.1 ± 127.3
Stocking density, animals/ha/21 d								
Cows (563 kg)	11.7	10.3	10.1	10.9	11.4	11.4	12.2	11.1 ± 1.1
450 kg BW	14.7	12.9	12.6	13.6	14.3	14.2	15.3	13.9 ± 1.3
100 kg BW	66.0	58.2	56.9	61.2	64.4	64.1	68.7	62.5 ± 6.1
Pasture height, cm								
Pre-grazing	34.1 ± 10.4	37.7 ± 19.2	39.6 ± 12.3	35.8 ± 9.8	38.1 ± 12.9	40.6 ± 13.4	35.4 ± 8.3	37.7 ± 3.4
Post-grazing	32.9 ± 9.0	28.1 ± 8.5	31.7 ± 6.5	32.9 ± 7.0	31.6 ± 5.8	32.4 ± 7.6	29.5 ± 5.1	30.1 ± 3.0
Post-mowing	22.3 ± 3.1	25.4 ± 7.7	25.2 ± 6.5	25.3 ± 6.6	28.9 ± 12.1	27.3 ± 9.8	23.8 ± 4.9	24.3 ± 2.3
Pasture mass								
kg of dry matter/ha								
Pre-grazing	11,934 ± 1993	12,628 ± 3679	12,981 ± 2359	12,248 ± 1874	12,701 ± 2472	13,170 ± 2568	12,182 ± 1587	12,622 ± 646
Post-grazing	11,699 ± 1724	10,774 ± 1638	11,467 ± 1244	11,699 ± 1348	11,458 ± 1114	11,607 ± 1452	11,052 ± 970	11,172 ± 575
Post-mowing	9666 ± 586	10,255 ± 1468	10,226 ± 1254	10,242 ± 1275	10,927 ± 2315	10,622 ± 1883	10,606 ± 1691	9068 ± 993
kg of dry matter/paddock								
Pre-grazing	1018 ± 170	1222 ± 356	1285 ± 234	1127 ± 172	1110 ± 216	1158 ± 226	998 ± 130	1150 ± 151
Post-grazing	998 ± 147	1043 ± 159	1135 ± 123	1076 ± 124	1002 ± 97	1020 ± 128	905 ± 79	1016 ± 118
Post-mowing	825 ± 50	993 ± 142	1012 ± 124	942 ± 117	955 ± 202	934 ± 166	869 ± 54	824 ± 116
Forage allowance, kg of dry matter/d								
Per cow (563 kg)	48.5 ± 8.1	58.2 ± 17.0	61.2 ± 11.1	53.7 ± 8.2	52.9 ± 10.3	55.1 ± 10.8	47.5 ± 6.2	54.8 ± 7.2
Per 450 kg BW	38.7 ± 6.5	46.5 ± 13.6	48.9 ± 8.9	42.9 ± 6.6	42.2 ± 8.2	44.1 ± 8.6	38.0 ± 4.9	43.8 ± 5.8
Per 100 kg BW	8.6 ± 1.4	10.3 ± 3.0	10.9 ± 2.0	9.5 ± 1.5	9.4 ± 1.8	9.8 ± 1.9	8.4 ± 1.1	9.7 ± 1.3
Pasture intake, kg of dry matter/d								
Per cow	2.0 ± 2.3	8.5 ± 10.1	7.1 ± 6.9	2.4 ± 2.3	8.3 ± 7.5	6.5 ± 4.9	4.4 ± 2.5	7.5 ± 3.9
Per 450 kg BW	1.6 ± 1.8	6.8 ± 8.1	5.7 ± 5.5	1.9 ± 1.9	6.6 ± 6.0	5.2 ± 3.9	3.5 ± 2.0	6.0 ± 3.2
Per 100 kg BW	0.4 ± 0.4	1.5 ± 1.8	1.3 ± 1.2	0.4 ± 0.4	1.5 ± 1.3	1.2 ± 0.9	0.8 ± 0.4	1.3 ± 0.7
Volumetric density, kg of dry matter/m^3^								
Pre-grazing	3.6 ± 0.6	3.7 ± 0.9	3.4 ± 0.4	3.5 ± 0.5	3.4 ± 0.4	3.3 ± 0.5	3.5 ± 0.4	3.5 ± 0.1
Post-grazing	3.7 ± 0.6	4.0 ± 0.7	3.7 ± 0.4	3.6 ± 0.4	3.7 ± 0.4	3.6 ± 0.4	3.8 ± 0.3	3.8 ± 0.2
Post-mowing	4.4 ± 0.3	4.1 ± 0.7	4.1 ± 0.6	4.1 ± 0.6	4.0 ± 0.9	4.0 ± 0.8	3.9 ± 0.7	4.2 ± 0.2
Accumulation ^2^, kg of dry matter								
Per ha	1338 ± 1072	1046 ± 821	1464 ± 182	1153 ± 356	1608 ± 542	1173 ± 529	3352 ± 2760	1556 ± 761
Per paddock	114 ± 91	101 ± 79	145 ± 18	106 ± 33	141 ± 47	103 ± 47	275 ± 226	140 ± 66
Pasture intake, kg of dry matter								
Per ha	501 ± 556	1854 ± 2190	1514 ± 1463	549 ± 534	1994 ± 1810	1564 ±1164	1130 ± 646	1639 ± 772
Per paddock	43 ± 48	180 ± 212	150 ± 145	51 ± 49	174 ± 158	137 ± 102	93 ± 53	157 ± 83
Grazing efficiency, %								
Intake/Mass	3.8 ± 3.8	12.3 ± 12.7	10.8 ± 8.7	4.1 ± 3.8	13.8 ± 10.3	11.0 ± 7.0	8.9 ± 4.3	12.1 ± 5.1
Intake/Accumulation	18.5 ± 11.6	103.7 ± 101.2	45.4 ± 14.2	20.4 ± 26.4	44.4 ± 13.5	115.5 ± 131.1	48.1 ± 56.3	95.6 ± 61.0

^1^ Area per 21 cows of 563 kg BW, per 26 animal units of 450 kg, or per 118 animals of 100 kg. ^2^ Average of the pre-grazing measurements minus the post-mowed measurements from the previous experimental period.

**Table 2 animals-13-01932-t002:** Composition of feed ingredients (mean ± SD of composite samples per period, *n* = 3).

Item ^1^	% of As-Fed	% of Dry Matter						
	DM	CP	NDF	EE	Ash	NFC	uNDF	Starch
Pasture ^2^	22.7 ± 0.6	12.7 ± 1.0	56.8 ± 3.0	2.1 ± 0.2	9.2 ± 0.7	19.2 ± 3.1	13.5 ± 0.8	3.2 ± 0.9
Corn silage	33.2 ± 1.7	6.4 ± 1.4	42.2 ± 4.3	3.4 ± 0.3	4.5 ± 0.7	43.7 ± 2.6	16.3 ± 0.2	18.4 ± 0.8
Soybean meal	89.6 ± 0.6	46.9 ± 0.3	10.9 ± 0.3	2.8 ± 0.2	5.6 ± 0.2	33.8 ± 0.7	4.4 ± 0.5	1.7 ± 0.4
Ground corn	88.6 ± 1.8	7.6 ± 0.3	8.2 ± 0.4	4.2 ± 0.2	1.3 ± 0.6	78.7 ± 0.2	3.8 ± 0.2	58.4 ± 4.6
Ensiled fine corn	63.1 ± 1.6	9.4 ± 0.6	9.9 ± 1.0	2.8 ± 0.2	1.4 ± 0.1	76.5 ± 1.7	4.4 ± 0.6	61.7± 4.3
Ensiled coarse corn	63.9 ± 1.0	9.1 ± 0.5	9.9 ± 0.3	2.8 ± 0.1	1.4 ± 0.1	76.8 ± 0.3	4.2 ± 0.2	64.6 ± 3.0

^1^ DM = dry matter. CP = crude protein. NDF = neutral detergent fiber. EE = ether extract. NFC = non-fiber carbohydrates = 100 − (CP + NDF + EE + ash). uNDF = undigested neutral detergent fiber (240 h in situ incubation). ^2^ Pasture DM and CP were analyzed from samples collected by hand plucking and other nutrients were analyzed from samples collected with an evacuated rumen-cannulated cow.

**Table 3 animals-13-01932-t003:** Pasture and total dry matter intake (DMI), digestible organic matter intake (DOMI), lactation performance, milk urea-N (MUN), linear somatic cell count (Ln SCC), and feed efficiency of high, medium, and low groups (two squares, each with three cows per group) fed with ground corn (GC), ensiled fine corn (FI), and ensiled coarse corn (CO).

Item	Treatment			SEM ^1^	Group			SEM ^1^	*p*-Value		
	GC	FI	CO		High	Medium	Low		Treat	Group	Treat × Group
	kg/d										
Pasture intake	7.5	6.3	7.2	0.54	8.0	6.7	6.2	0.70	0.16	0.20	0.54
DMI	18.1 ^a^	16.7 ^b^	17.7 ^ab^	0.54	19.7 ^a^	17.2 ^b^	15.5 ^b^	0.70	0.09	<0.01	0.53
DOMI	10.6 ^a^	9.6 ^b^	9.9 ^ab^	0.40	11.3 ^a^	10.0 ^b^	8.7 ^c^	0.53	0.10	0.01	0.28
Milk	18.9	19.4	19.4	0.73	25.4 ^a^	17.7 ^b^	14.5 ^b^	1.15	0.54	<0.01	0.47
ECM ^2^	16.6	16.9	17.0	0.62	21.0 ^a^	16.0 ^b^	13.4 ^b^	0.93	0.77	<0.01	0.83
4% FCM ^3^	16.0	16.2	16.4	0.61	20.3 ^a^	15.3 ^b^	12.9 ^b^	0.91	0.73	<0.01	0.87
Fat	0.552	0.564	0.576	0.0267	0.680 ^a^	0.548 ^b^	0.464 ^b^	0.0398	0.60	<0.01	0.97
Protein	0.561	0.576	0.569	0.0194	0.702 ^a^	0.551 ^b^	0.453 ^c^	0.0302	0.61	<0.01	0.29
Lactose	0.840	0.868	0.862	0.0337	1.147 ^a^	0.787 ^a^	0.637 ^b^	0.0528	0.50	<0.01	0.62
Total solids	2.134	2.171	2.195	0.0770	2.761 ^a^	2.053 ^b^	1.686 ^c^	0.1199	0.59	<0.01	0.42
	%										
Fat	2.99	2.94	3.05	0.121	2.68	3.06	3.24	0.195	0.33	0.15	0.79
Protein	3.05	3.12	3.00	0.097	2.78 ^b^	3.22 ^a^	3.17 ^ab^	0.146	0.37	0.10	0.57
Lactose	4.43	4.44	4.43	0.051	4.51	4.42	4.37	0.077	0.98	0.44	0.88
Total solids	11.43	11.34	11.45	0.194	10.90	11.64	11.69	0.330	0.17	0.19	0.80
MUN, mg/dL	13.5	14.6	14.0	0.79	13.7	15.6	12.8	1.12	0.44	0.24	0.12
Ln SCC, ^4^ 1 to 9	3.03	3.37	3.36	0.318	3.07	4.03	2.65	0.496	0.32	0.16	0.26
Milk/DMI	1.03 ^b^	1.18 ^a^	1.08 ^b^	0.050	1.31 ^a^	1.04 ^b^	0.94 ^b^	0.078	<0.01	0.01	0.97
Milk/DOMI	1.78 ^b^	2.12 ^a^	1.95 ^ab^	0.117	2.27 ^a^	1.72 ^b^	1.64 ^b^	0.133	0.02	0.06	0.97
ECM/DMI	0.90 ^b^	1.03 ^a^	0.95 ^b^	0.045	1.09 ^a^	0.94 ^b^	0.86 ^b^	0.069	<0.01	0.09	0.97
ECM/DOMI	1.55 ^b^	1.86 ^a^	1.72 ^ab^	0.079	1.89 ^a^	1.54 ^b^	1.50 ^b^	0.123	0.05	0.08	0.94

^1^ Standard error of the means. ^2^ ECM = energy-corrected milk. ^3^ 4% FCM = fat-corrected milk. ^4^ Equivalency of the linear SCC: 3.03 = 102.000 cells/mL, 3.37 = 129.000 cells/mL, 3.36 = 128.000 cells/mL, 3.07 = 105.000 cells/mL, 4.03 = 204.000 cells/mL, and 2.65 = 78.5000 cells/mL. ^a–c^ Means in a row within factor with differing superscripts differ at *p* ≤ 0.05 by Tukey–Kramer.

**Table 4 animals-13-01932-t004:** Ingredient and nutrient composition (% of dry matter) of the consumed diets fed to high-, medium-, and low-producing cows fed with ground corn (GC), ensiled fine corn (FI), and ensiled coarse corn (CO).

Item	High			Medium			Low		
	GC	FI	CO	GC	FI	CO	GC	FI	CO
Pasture	40.8	38.7	42.6	42.6	33.6	38.5	39.8	39.5	40.4
Corn silage	13.5	14.2	13.3	14.4	16.9	15.6	17.2	17.6	17.2
Soybean meal	13.4	14.2	13.3	12.9	15.1	14.0	12.0	12.2	12.0
Ground corn	30.5			28.3			28.8		
Ensiled fine corn		31.1			32.1			28.4	
Ensiled coarse corn			29.1			29.8			28.1
Premix ^1^	1.8	1.9	1.8	1.9	2.3	2.1	2.3	2.3	2.3
Crude protein	14.7	15.4	15.1	14.5	15.5	15.1	13.9	14.5	14.4
Neutral detergent fiber (NDF)	32.8	32.6	34.1	34.0	31.0	32.9	33.5	34.0	34.3
Ether extract	3.0	2.6	2.5	2.9	2.6	2.6	3.0	2.6	2.5
Ash	7.3	7.3	7.4	7.6	7.4	7.5	7.8	7.8	7.9
Non-fiber carbohydrates ^2^	42.3	42.2	40.8	41.1	43.5	41.8	41.8	41.1	40.9
Starch	21.8	23.3	22.9	20.8	24.3	23.6	21.5	22.2	22.9
Pasture NDF	23.2	22.0	24.2	24.2	19.1	21.9	22.6	22.4	22.9
Silage NDF	5.7	6.0	5.6	6.1	7.1	6.6	7.2	7.4	7.3
Forage NDF	28.9	27.9	29.8	30.2	26.2	28.4	29.8	29.8	30.2

^1^ 19.0% Ca; 6.0% P; 4.3% Mg; 0.2% S; 15 mg/kg Co; 700 mg/kg Cu; 3730 mg/kg Mn; 2500 mg/kg Zn; 19 mg/kg Se; 40 mg/kg I; 200k IU/kg vitamin A; 50k IU/kg vitamin D; and 1500 IU/kg vitamin E. ^2^ NFC = 100 − (CP + NDF + ether extract + ash).

**Table 5 animals-13-01932-t005:** Particle size distribution, geometric mean particle size (GMPS), particles per gram, surface area, ruminal in situ dry matter degradation over time (Deg), effective ruminal degradation (ERD), fractional degradation rate (kd), silage pH, and dry matter concentration (DM) of ground corn (GC, *n* = 9), fine corn (Fine) and coarse corn (Coarse) before and after ensiling (*n* = 18/treatment).

Item	GC	Before Ensiling		After Ensiling		SEM ^1^	*p*-Value
		Fine	Coarse	Fine	Coarse		
	Particle size						
	% of particles on sieve µm						
4750	0.0 ^c^	1.6 ^ab^	1.3 ^b^	1.9 ^a^	2.0 ^a^	0.82	<0.01
2830	0.0 ^c^	3.1 ^b^	14.0 ^a^	3.3 ^b^	14.3 ^a^	0.58	<0.01
1400	1.5 ^e^	47.2 ^c^	53.5 ^a^	41.0 ^d^	50.0 ^b^	0.82	<0.01
1000	5.6 ^c^	20.0 ^a^	12.5 ^b^	19.8 ^a^	12.5 ^b^	0.37	<0.01
600	26.7 ^a^	18.6 ^b^	11.9 ^c^	20.9 ^b^	12.9 ^c^	0.73	<0.01
250	33.6 ^a^	7.4 ^c^	5.1 ^d^	10.3 ^b^	6.2 ^cd^	0.58	<0.01
Pan	32.7 ^a^	2.1 ^Bc^	1.7 ^c^	2.7 ^bA^	2.0 ^c^	0.18	<0.01
GMPS ^2^, µm	366 ^d^	1364 ^b^	1694 ^a^	1258 ^c^	1648 ^a^	25.7	<0.01
Particles ^2^, /g	318,308 ^a^	2086 ^b^	1083 ^b^	3652 ^b^	1520 ^b^	1643	<0.01
Surface area ^2^, cm^2^/g	45.0 ^a^	27.3 ^c^	24.6 ^d^	28.7 ^b^	25.3 ^d^	0.26	<0.01
	Ruminal degradation						
	% of dry matter						
Deg 0 h	21.0 ^a^	15.3 ^b^	13.0 ^b^	23.2 ^a^	22.6 ^a^	1.10	<0.01
Deg 3 h	25.4	19.0	18.2	23.8	22.5	1.99	0.17
Deg 6 h	27.2 ^ab^	23.6 ^b^	23.7 ^b^	31.2 ^a^	28.8 ^ab^	1.99	<0.01
Deg 18 h	45.1 ^ab^	36.6 ^bc^	34.3 ^c^	49.3 ^a^	47.8 ^a^	1.76	<0.01
Deg 48 h	79.5 ^ab^	75.5 ^bc^	71.9 ^c^	83.6 ^a^	79.2 ^ab^	1.66	<0.01
ERD ^3^	42.7 ^ab^	37.6 ^bc^	34.5 ^c^	46.2 ^a^	44.0 ^a^	1.12	<0.01
kd of fraction B, %/h	2.53 ^abc^	2.34 ^bc^	2.12 ^c^	2.86 ^a^	2.54 ^ab^	0.119	<0.01
pH				3.89	3.94	0.02	0.17
DM, % of as-fed		63.4	63.0	63.7	63.3	0.20	0.16

^1^ Standard error of the means. ^2^ Kansas State University: MF-2051. Baker and Herrman (2002). ^3^ Effective ruminal degradation = A + B × [kd/(kd + kp)]. kp = fractional passage rate (6.5%/h). ^a–e^ Means in a row with differing superscripts differ at *p* ≤ 0.05 by Tukey–Kramer. A–B Means in a row with differing superscripts differ at *p* ≤ 0.10 by Tukey–Kramer.

**Table 6 animals-13-01932-t006:** Total tract apparent digestibility of dry matter (DM), organic matter (OM), neutral detergent fiber (NDF), starch, fecal viscosity, and starch concentration, as well as plasma urea-N (PUN) and glucose concentrations of high, medium, and low groups (two squares, each with three cows per group) fed with ground corn (GC), ensiled fine corn (FI), and ensiled coarse corn (CO).

Item	Treatment			SEM ^1^	Group			SEM ^1^	*p*-Value ^2^		
	GC	FI	CO		High	Medium	Low		Treat	Group	Treat × Group
Digestibility, % of intake											
DM	57.5 ^a^	58.1 ^a^	55.7 ^b^	0.66	57.1	57.8	56.4	0.67	0.03	0.34	0.77
OM	60.7 ^a^	61.0 ^a^	58.6 ^b^	0.64	59.9	61.0	59.5	0.74	0.01	0.36	0.25
NDF	45.6 ^b^	48.9 ^a^	45.9 ^ab^	1.17	47.5	46.5	46.4	1.22	0.09	0.81	0.51
Starch	92.6 ^a^	92.0 ^a^	86.3 ^b^	1.04	87.8 ^b^	91.6 ^a^	91.6 ^a^	1.04	<0.01	0.01	0.16
Fecal viscosity, cP	33.4	36.2	33.2	2.39	37.5	33.0	32.4	2.39	0.61	0.27	0.34
Fecal starch, % of DM	3.6 ^b^	4.4 ^b^	7.0 ^a^	0.49	6.4 ^a^	4.5 ^b^	4.1 ^b^	0.49	<0.01	<0.01	0.18
Glucose, mg/dL	70.8 ^b^	75.7 ^a^	74.3 ^a^	1.00	73.1	74.7	72.9	1.13	<0.01	0.49	0.13
PUN, mg/dL	22.0	21.7	22.2	1.01	22.2	22.4	21.3	1.01	0.93	0.71	0.75

^1^ Standard error of the means. ^2^ Glucose and PUN: samples obtained at 0500, 1000, and 1400 h. *p* ≥ 0.26 for time and *p ≥* 0.27 for the interaction between treatment, group, and time. ^a–b^ Means in a row within factor with differing superscripts differ at *p* ≤ 0.05 by Tukey–Kramer.

**Table 7 animals-13-01932-t007:** Ruminal volatile fatty acids (VFA), pH, and ammonia-N after feeding with ground corn (GC), ensiled fine corn (FI), and ensiled coarse corn (CO) (data from one square of three rumen-cannulated cows). Sampling times were 0500, 1000, and 1400 h.

Item	Treatments			SEM ^1^	*p*-Value		
	GC	FI	CO		Treat	Time	Treat × Time
VFA, mM							
Acetate	64.4	62.2	64.7	4.37	0.88	0.36	0.51
Propionate	26.6	25.2	28.4	4.23	0.72	0.22	0.77
Butyrate	9.8	10.1	9.9	0.50	0.96	0.34	0.52
Isobutyrate	1.1	1.1	1.1	0.09	0.99	0.38	0.52
Valerate	1.3	1.3	1.8	0.14	0.29	0.36	0.80
Isovalerate	2.0	2.1	2.3	0.17	0.86	0.46	0.51
Total VFA	105.3	102.2	108.1	6.82	0.81	0.32	0.57
VFA, % of total molar							
Acetate	61.4	61.7	61.2	2.64	0.90	<0.01	0.77
Propionate	25.0	23.9	24.9	2.75	0.73	0.01	0.99
Butyrate	9.3	10.0	9.3	0.53	0.62	0.36	0.96
Isobutyrate	1.1	1.1	1.1	0.06	0.88	<0.01	0.89
Valerate	1.3	1.3	1.5	0.24	0.58	0.97	0.93
Isovalerate	2.0	2.1	2.1	0.15	0.78	0.12	0.78
Acetate/Propionate	2.52	2.66	2.72	0.359	0.22	<0.01	0.73
pH	6.06	6.13	6.06	0.16	0.99	<0.01	0.58
Ammonia-N, mg/dL	16.0	15.2	15.1	4.15	0.52	0.37	0.45

^1^ Standard error of the means.

## Data Availability

Not applicable.

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
