# Peer review of "Effect of Particle Size of Silage of Flint Corn Grain on Dairy Cows Fed Tropical Pasture: Performance, Intake, Ruminal Fermentation, and Digestibility"

_animals, 2023, doi:10.3390/ani13121932_

Round 1
Reviewer 1 Report
All opinion was emitted with all respect to the efforts of the authors for the preparation of the experiment and its report
The purpose of this study was to evaluate the effect of the particle size of reconstituted and ensiled corn grain offered to grazing high, medium, and low milk yield dairy cows under subtropical climate on nutrient digestion, ruminal fermentation, milk yield, and blood metabolites. The manuscript provides interesting information on the subject. However, few details must be corrected in order for this paper to be published.
General
Redaction in the whole document is very hard to follow. This is made worse because the document is not properly structured, and there is a large amount of data presented that is unnecessary, or has very little relation to the objective of the study.
Title: Is vague and imprecise.
Must be improve (Hint) Effect of particle size of reconstituted and ensiled corn grain offered to grazing high, medium, and low milk yield dairy cows under subtropical climate: nutrient digestion, ruminal fermentation, and milk yield
Simple summary: Too long, shorten please.
L10: Corn grain
L10-11 The most common term to describe the cold process of cereal grains by rehydration and ensiled is "reconstitution". Please, apply this term here and through the document. Rewrite as (Hint): The corn grain commonly used in Brazil is a hybrid of high vitreousness that has a lower starch digestibility, in such a manner that processing of corn grain before offered to cattle is recommended. The cold process of cereal grains by rehydration and ensiled (reconstitution) is a low-cost way to store corn grain and can increase starch digestibility.
Abstract
Interactions (processing x group production) are not exposed in abstract. Please include
L24-25: on pasture dry matter intake (DMI),
L27: Rewrite: Eighteen Holstein cows (average daily milk yield =21.3 kg) were blocked (3 blocks) by milk yield, and were..
L28: Lack of clarity. The fixed quantity of 2.7 kg was of whole plant silage + corn + soybean meal? Or only the whole plant? Please, specify clearly the fixed amount of whole plant, of the concentrate and the proportion of corn and soybean meal in the concentrate.
L29: Since concentrate plus whole plant were offered at constant amounts; then, increases or decreases on DMI only obey by changes on pasture intake. You should clarify this to the readers.
L30: Write as: There were no treatment effects on milk yield (average=19.2 kg/d) and energy corrected..
L31: Include treatments effect on starch digestion (important variable in your study!). Moreover, write as NDF digestibility was lower for GC and CO than FI.
L33: Conclusion must be improved (Hint): Fine grinding (3 mm sieve) of reconstituted corn grain used as supplement at level of (?) reduced pasture intake in grazing cows without affecting milk yield improving feed efficiency. Even when grinding reconstituted grain coarsely (8 mm sieve) significantly reduced total tract starch digestion did not significantly affect feed efficiency. Reconstituted grain corn and grind before feeding increase the energy value compared to dry grind corn.
Introduction
There are a lot of concepts that are leftover and must be removed. Authors should have better focus on the concepts that sustain the rationale of treatment related to the main objective of the experiment.
L45-50: Redaction very difficult to follow!! Please rewrite as (Hint): The response in DMI, lactation performance and rumen fermentation of grazing cows to the energy supplementation with cereals grain may be affected by the source of grain and type of processing thereof [6], amount and frequency of supplementation [7,8], and by the method and strategy of supplementation [9-14].
L51: Compared to concentrate supplemented alone ?? Confuse statement (rewrite)
L53: Could be? reduce peak of concentrate intake lowering ruminal NH3-N and risk of lower ruminal pH
L55: with a controlled environment? Or in a shaded facility?
L58: Removed worldwide, is leftover.
L91: Supplemented at fixed quantity? Level of supplementation related to pasture? Please, specify.
Material and methods
It is redacted in a disorderly way. It is difficult to follow the reading, it is tiring and confusing as it is presented. Must be improved
Describe first the conditions generals that all cows were subjected, then describe specific methods of each trial performed (milk performance trial, in vivo digestion and fermentation trial, and ruminal in situ degradation) in this sense reorder subheading as follows:
2.1 Farm location and climatic conditions
2.2 Pasture management
2.2 Corn grain processing (should include of dry corn (GC) processing as well)
2.3. Dietary treatments
2.4. Milk performance trial
2.4.1. Experimental unit characteristics, housing and feed management, duration of the trial, sampling methods, sample analyses, and calculations
2.5. In vivo digestion trial
2.5.1. Experimental unit characteristics, housing and feed management, duration of the trial, sampling methods, sample analyses, and calculations
2.6. Ruminal in situ degradation
2.6.1. Experimental unit characteristics, housing and feed management, duration of the trial, sampling methods, sample analyses, and calculations
2.7. Statistical analyzes
Other observations
L97-107 (2.1 Farm location and environment): Too much information that is leftover in this subheading. Please, simplify as follows (Hint): The experiment was conducted at the Três Pontas experimental farm of the Agricultural Research Company of Minas Gerais (Epamig), in Três Pontas, MG, Brazil (21° 20’ 99 25.92” S, 45° 28’ 46.01” W), from January 15 to March 18, 2018. The farm is located at 100 m above sea level and has a humid subtropical climate Cwa [XX]. During the course of the experiment ambient air temperature averaged 22.3 ± 3.6 °C (minimum and maximum of, respectively), and relative humidity averaged 76.0 ± 7.8% (minimum and maximum, respectively). Thus, according to Youself [35], the estimated temperature-humidity Index (THI) averaged 69.8 ± 3.6 (minimum and maximum, respectively). END
L109-114: Remove Figure 1 and 2.
L128: It is essential to minimize the use of unnecessary acronyms. Please, describe as “corn grain”, remove MCG” and “MRE” Use comma instead dot (GC, 1.5 m sieve)
L131: Need a better description!! (Hint)..and were individually fed with supplements three times daily without feed refusals (equal quantities). Concentrate supplement was composed of corn grain and soybean meal in a proportion of XX:XX, respectively, and was offered XXX kg/d distributed at schedule feeding of 0500, 1000, and 1400 h. A total of 4 kg (as-fed basis) of whole plant silage was mixed with the concentrate supplement and was offered at 1000, and 1400 h feeding. Chemical composition of supplement ingredients is shown in Table XX.
L169: How much time elapsed before the reconstituted corn finished ensiling was processed? How much DM contained reconstituted corn at time of processing? The processed grain was immediately offered to cows? Or was stored by a time before the offer? Please specify.
L225-226: Diets were prepared daily, weekly, monthly?? Specify
L227-239: Too many words! The main objective was to evaluate the effect of particle size of reconstituted corn grain offered as a supplement in grazing cows. Focus on this point (corn grain)! Authors could indicate here (Hint): Quantity of supplemented corn grain (not of supplement) in each milk yield category were 6.1, 5.3, and 4.5 kg/d and were determined considered a fixed offered of 2.8 kg/d of supplemental whole corn plant silage, and in estimated pasture (grazing) DMI of 20.4, 18.6, 16.6 kg for high, medium, and low milk yield groups [41]. Ingredients and chemical characteristics of plant, whole plant corn silage, and corn grain are shown in Table XX. END
L250: fecal output
L251: According to the description the external marker offered was 3 g Cr2O3 per cow/feeding, then a total of 9 g Cr2o3 cow/day? Please describe as: fecal output was estimated using Cr2O3 as an external marker dosed at a total of 9 g daily/cow. The marker was mixed.
Please, indicate the purity of Cr2O3 used.
L286. To an adapted protocol from cannon et al. [47]; briefly, fecal samples were..
Statistical analyses
For milk yield trial, you use a replicated 3 x 3 Latin square (6 replicas), but for in vivo digestion trial a unreplicated 3x3 Latin square design was used. At present it is difficult to accept digestion trials performed as an unreplicated 3x3 LS (low statistical power). Authors should mention this limitation to the readers.
Results
L339-340: Indicate code description for THI values estimated by Youlsef formula, i.e. THI code (Normal THI < 74; Alert >74-79; Danger 79-84 and Emergency > 84)
L342: The frequency of THI ≥ 68 was 69% of the time and ≥ 72 was 65% of the time. How can this be? A better description is regard to hours/day. For example, cows were in an environment of THI >77 an average of 6-h daily during the experiment.
L345: Table 1 should be removed; you can send this information (and other removed) as additional (complementary) files.
L355-369: Please rewrite as (Hint): Chemical composition of pasture, as well as ingredients used as supplement (whole plant corn silage, corn grain dry-ground and reconstituted, and soybean meal) is shown in Table 1. The NDF concentration evaluated using simulated grazing were slightly lower (5.3%, 53.8 vs 56.8%) than nutrient concentration by samples obtained in rumen cannulated cows. Both values were lower than the expected NDF concentration (63%) estimated by diet composition (NRC, 2007). The DM content of REC was close to the planned (63.5 vs 65%). The uNDF concentration determined by the sample from cannulated cows and by simulated grazing was very similar (13.5 vs 11.8%). The correlation of uNDF concentration measured by the two sampling procedures was high (r2 = 0.99) and uNDF was consistently higher on samples harvested with a cow than by simulated grazing. The uNDF concentration was higher during period 1 than in periods 2 and 3, denoting the positive relationship between pasture allowance and uNDF concentration. END.
L367: remove figure 5 from document. Send as a complementary file
L386-387: What is this? Please rewrite it clearly. You can start this section as (Hint): Estimates of pasture intake obtained by both sampling procedures and marker data per cow were close to the mean pasture intake estimated from pre-grazing -post-grazing pasture height. There was no treatment..
L392: (averaged 7 kg DM)
L394: 9 diets!! This experiment tested dietary treatments. Your design is a replicated 3x3 LSQ, then, you only have 3 dietary treatments. Or were insulated Latin squared? Don't tread on quicksand! Anyway, the paragraph is confusing, clarify. (I recommend indicate as experimental diets (remove 9)
L394: You need to be clearer. Total forage is pasture plus whole corn plant silage? If yes, then, specify it
L497: Ruminal pH was lowest and the acetate: propionate ratio was highest in samples taken at 0500
Discussion
Discussion is well performed only few flaws
L579: The concentration of 23% of starch, per se, is not a high starch quantity to reduce dramatically ruminal pH, but the ruminal digestion rate (velocity) of the starch is. Then, apparently, compared to dry-grinding, reconstitution did not increase ruminal starch digestion rate in this experiment. Please, consult Zebeli et al. 2008. JDS.91:2046.
L591: As exposed above, unreplicated 3x3 LS have a low statistical power, moreover, normally, NDF determination has a high variation, thus, this result can be more by experimental design than for treatment effect. This must be exposed to the readers.
Conclusions
Please reorder, start with (Hint): There were no interactions between processing corns and production level. Particle size of REC did not affect DMI, but coarse grinding reduced total tract starch digestibility without effect on milk yield, ruminal fermentation parameters, nor blood metabolites. Cows that were feeding with fine grinding of dry corn grain (GC) increased feed efficiency, compared with CO, but was similar to FI. Particle size of REC showed a similar ruminal pH compared to the cows fed finely grinding corn.
L610: milk components, but induced low ruminal pH
Author Response
Please see the attachment. Very thanks.

Reviewer 2 Report
This manuscript presents an original study that aimed to study the effect of corn grinding and ensiling on the performance of lactating dairy cows. The study is comprehensive, well-controlled, and has many analyses. However, it has a couple of weaknesses: 1) there is a potential confounding effect of corn particle size and ensiling, especially when treatment GC is compared with FI and CO; 2) the discussion section is very weak and it sounds more like a results section. Please, read the specific comments below.
L.13. Why are you talking about the storage period here? Did you evaluate the storage period?
L.13-15. Please review this sentence. It is a bit confusing.
L.25 – delete “(DM)”
L.142 – I suggest presenting the pasture management before study design.
L. 174 – Why GC corn was ground through a different screen?
L. 248 – It is more commonly referred to as iNDF.
L. 257 – Can you add a reference for the 240h of incubation?
L. 258-261 – Can you add a reference for this calculation? Were all this uNDF (or iNDF) expressed in g/d? Please explain it.
L. 300-301 – I suggest “samples were pooled per cow, sampling time, and period.” It is confusing to understand how you generated these pooled samples.
L. 320 – What do the “-Before” and “-After” mean?
L. 328- Was time here included as a repeated measurement?
L. 335 – I suggest removing this statistical analysis of the trend. It is so confusing to read your results in the tables. Also, why are you running an ANOVA if you are going to run a Tukey test regardless of the ANOVA P-value? You can discuss the trend, I don’t think it makes sense to show in tables both results.
L. 346 – Is this a daily allowance?
Tables 4 and 3 – It is weird that you are presenting ingredient composition as a result, and after showing intake and performance data.
L. 443-454 – This section should be concatenated with 3.2. It is weird that you are going back to Table 3.
L. 501-602 – The whole discussion section is repetitive and sounds more like a results section. You should really discuss your data here or combine it with the results. Either way, you should expand the discussion of your results, not only present them.
L. 504 – Is this recommendation of 15 animal units/ha for continuous grazing?
L. 515-517 – Please connect this sentence to your results or remove it.
L. 522 – Did you use Holstein cows?
L. 524-526 – How do your results support the sentence “Concentrate and forage supplementation occurred during the warmer hours of the day, and it seemed to be a reasonable strategy for Holstein cows grazing in a tropical environment”?
L. 527 – This should be described in the material and methods. If you are not discussing it, delete it from this section.
L. 541 – Why are you repeating this grazing efficiency here?
L. 552-555 – How is this hypothesis supported by your data? Ruminal propionate did not differ among treatments, did it?
L. 559-560 – It is hard to explain some of your results because you have a confounding effect of particle size and ensilaging, that is, you don’t know if the different result between GC and either FI or CO is a consequence of the ensiling or the particle size. I think you should make the reader aware of this confounding effect.
L. 564 – Was there a different source?
L. 565 – “ruminal” instead of “rumen”
L. 565-567- I think this is a strong sentence. Is it really true?
L. 582-584 – Are you speculating, or does your date support this affirmation?
L. 586 – Does “numerically lower” means different?
L. 585-588 – This sentence is wordy, and it does not make sense.
L.593 - What do you mean by “fiber-digesting bacteria of starch 593 hydrolysis products”?
L. 596-597 – Again, this is the discussion section, not the results.
Author Response
Please see the attachment. Very thanks.

Round 2
Reviewer 1 Report
I have read the revised manuscript and appreciate the authors' consideration of my previous suggestions. I have no further reviewer comments.